# Dynamic Observation Policies in Observation Cost-Sensitive Reinforcement Learning

## Abstract

Reinforcement learning (RL) has been shown to learn sophisticated control policies for complex tasks including games, robotics, heating and cooling systems and text generation. The action-perception cycle in RL, however, generally assumes that a measurement of the state of the environment is available at each time step without a cost. In applications such as materials design, deep-sea and planetary robot exploration and medicine, however, there can be a high cost associated with measuring, or even approximating, the state of the environment. In this paper, we survey the recently growing literature that adopts the perspective that an RL agent might not need, or even want, a costly measurement at each time step. Within this context, we propose the Deep Dynamic Multi-Step Observationless Agent (DMSOA), contrast it with the literature and empirically evaluate it on OpenAI gym and Atari Pong environments. Our results, show that DMSOA learns a better policy with fewer decision steps and measurements than the considered alternative from the literature.

## 1   Introduction

In many applications of reinforcement learning (RL), such as materials design, computational chemistry, deep-sea and planetary robot exploration and medicine (13; 15; 2), there is a high cost associated with measuring, or even approximating, the state of the environment. Thus, the RL system as a whole faces observation costs in the environment, along with processing and decision making costs in the agent. On both sides, the costs result from a cacophony of factors including the use of energy, systems and human resources. In this work, we propose the Deep Dynamic Multi-Step Observationless Agent (DMSOA), the first RL agent in its class to reduce both measurement and decision making costs.

Since standard RL agents require a large number of *state-action-reward-next state* interactions with the environment during policy learning and application, the measurement and decision making costs can be very high. Traditionally, these underlying costs are hidden from the agent. Indeed, little consideration has been given to the idea that the agent might not need, or even want, a potentially costly observation at each time step. In the real-world, however, agents (animal or artificial) are limited by their resources. To save time and energy, decision making associated with common or predictable tasks is believed to be conducted re-actively or based on fast, low-resource systems. Only with deliberate cognitive intervention are the slower, resource-intensive planning systems used (22; 8).

Recently, there have been a number of interesting conference papers, workshop discussions and theses discussing how to address this problem in RL (4; 15; 5; 11; 6; 9; 20). The general approach is to augment RL by *a*) assign an intrinsic cost to measure the state of the environment, and *b*) provide

Submitted to the Workshop on Advancing Neural Network Training at 37th Conference on Neural Information Processing Systems (WANT@NeurIPS 2023). Do not distribute.

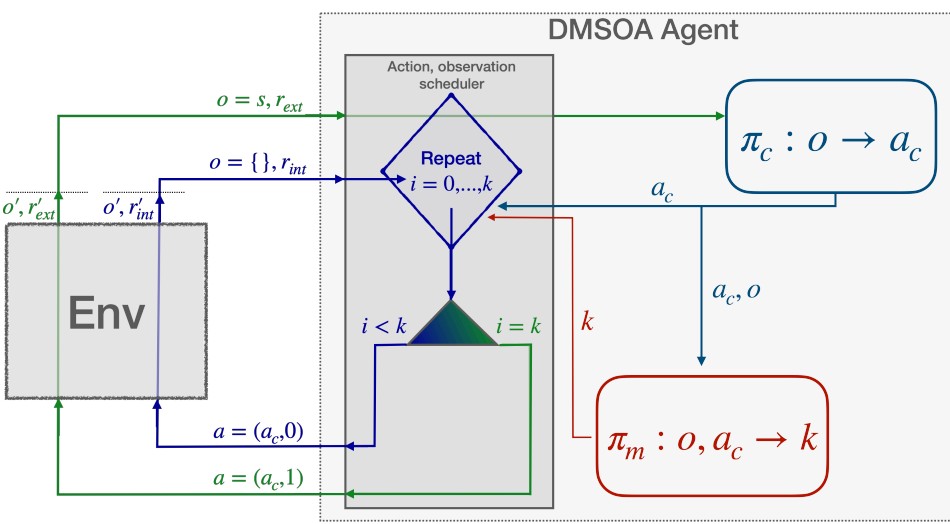

Figure 1: Illustration of the DMSOA Framework.

the agent with the flexibility to decide if the next state should be measured. Together, these provide a mechanism and a learning signal to encourage the agent to reduce its intrinsic measurement costs relative the to the explicit control rewards it receives.

When the agent opts not to measure the state of the environment, it must make its next control decision based on stale information or an estimate. Thus, at the highest level, this constitutes a partially observable Markov decision process (POMDP). Learning a POMDP, however, is much more difficult than learning a Markov decision process (MDP) with RL (17). When designed as shown in Fig. 1, the agent's experience is composed of fully observable measurements and partially observable estimates of the state of the environment. Thus, the problem is related to mixed observable Markov decision processes (MOMDPs), which are an easier sub-class of POMDPs (16). The authors in (15) denoted this class of RL problem as action-contingent, noiselessly observable Markov decision processes (AC-NOMDPs). Distinct from an MOMDP, action-contingent in AC-NOMDP relates the fact that the agent explicitly chooses between measuring and not measuring the environment. "Noiselessly observable" relates to the fact that when the agent decides to measure at a cost, the state is fully observable. Although previous works have used different terms to refer to this class of problem, we believe that AC-NOMDP is the most descriptive of the underlying dynamics and use it throughout this paper.

Recently, (15) provided a theoretical analysis of the advantages of AC-NOMDP of over a general POMDP formulation and found a significant improvement in efficiency for RL with explicit observation costs and actions. All other previous works have carried out limited empirical evaluations in which the proposed algorithm is compared to a baseline MDP (11; 5; 9; 20). Moreover, the previous analyses primarily relied on just a few, or even single, experiments on OpenAI gym classic control environments and grid-worlds (11; 5; 20).

In this work, we compare DMSOA to the one-step memory-based observationless agent (OSMBOA) recently proposed in (5). Our contribution serves to expand the state-of-the-art in AC-NOMDP algorithms and the understanding of how different classes of algorithms impact the observation behaviour. OSMBOA was selected for its demonstrated effectiveness and easy of use. At each time step, OSMBOA selects a control action and makes a decision about measuring the next state of the environment. If no measurement is made, the agent's next control action is selected based on its fixed-size internal memory of the last measured state(s). In contrast, DMSOA selects a control action and the number of times to apply the action before measuring the next state. Therefore, DMSOA learns to reduce both observation costs and decision making costs by dynamically applying its control action multiple times.

To facilitate fair comparison, we implement both agents as double DQN (25) with prioritized experience replay (19). We evaluate the agents in terms of the accumulated extrinsic control reward and by the reduction in the number of observations and decision steps made on Atari Pong and

OpenAI gym (7). The results show that the proposed method learns a better control policy, requires fewer measurements of the environment and decision steps. Moreover, DMSOA has less variance across independent training runs.

The remainder of the paper is organized as follows. In the next section, Section 2, we outline the related work. Section 3 formalizes the AC-NOMDP problem and Section 4 presents the proposed algorithms. Section 5 provides the experimental setup. The results are shown in Section 6 and discussed in Section 7. Our concluding remarks are in Section 8.

## 2    Related Work

This work fits into a small but growing sub-area of RL in which observations are optional at each time step and have an explicit cost to the agent when they are made. In the subsection immediately below, we provide an overview of methods recently applied to AC-NOMDP. Following that, we discuss the literature directly related to the proposed DMSOA algorithm.

### 2.1    AC-NOMDP Methods

In the existing work, the authors in (4; 15; 20) proposed tabular Q-learning-based algorithms for AC-NOMDPs and (11; 15; 5; 9) proposed deep RL based methods. In (11), the authors modify TRPO, and in (15; 9), actor-critic frameworks with a recurrent neural networks are used. In (5), the authors provide a wrapper class that modifies the underlying environment by expanding the observation and action spaces to facilitate any off-the-shelf deep RL algorithm to work in the AC-NOMDP setting.

Through our analysis of the literature, we have identified 4 key questions addressed when developing for AC-NOMDPs. These are: *1)* the mechanism by which the agent expresses its desire not to measure, *2)* how the observation is supplemented when no measurement is made, *3)* how the agent is encouraged to reduce its reliance of costly measurements, and *4)* how the agent is constructed.

The most common way to handle question *1)* is by expanding the action space. In the case of discrete actions, (4; 15; 5; 9; 20) expanded the action space to action tuples: ⟨control actions⟩ × ⟨measure, don't measure⟩. Alternatively, in (11), the agent specifies the control action plus a sample purity value $q \in \mathbb{R}^1$, where a larger $q$ triggers a less accurate measurement with lower associated costs.

With respect to question *2)*, when the agent does not request a fresh measurement in (11; 4; 15; 9), the environment sends a Null state observation or an observation composed of zeros. In (4), the agent uses an internal statistical model to estimate the next state and in (15; 9) the agents utilize a deep recurrent networks for estimating belief states and encoded states, respectively. In (5), the agent utilizes a fixed-size memory of recent measurements when no measurement is made. To reduce partial observability, each observation is augmented with a flag indicating whether or not it is the result of a fresh measurement of the environment. Since the agent in (11) adjusts the noise level rather than turning on and off measurements, it makes its next action selection purely based on the noisy measurement returned.

Question *3)* relates to the rewards structure. This is generally divided into intrinsic rewards (or costs), which are used to encourage the agent to reduce its reliance on costly measurements and extrinsic rewards that push the agent to achieve the control objective. At each time step in (11; 4; 15; 9), an intrinsic cost is subtracted from the extrinsic reward if the agent measures the state. Alternatively, in (21) a positive intrinsic reward is added when the agent foregoes a measurement. A critical point that remains unclear in the literature is how to acquire the extrinsic reward when no measurement is made. In most cases, this is simply assumed to be available. We argue that if no measurement is made, the extrinsic reward cannot be known. As a result, in this work only the intrinsic portion of the reward is provided when no measurement is made.

The final question relates to the architecture of the agent. In (4; 15; 5), a single agent policy select both the control action and the measurement behaviour. Alternatively, in (11; 9; 20), separate policies are learned to determine the control actions and measurement actions. In addition, (4; 15; 9) learn models for estimating the next state.

## 2.2 Works Related to DMSOA

The proposal of (20) is most algorithmically related to the DMSOA. In it, the author demonstrated the potential of dynamic action repetition for RL with observation costs using tabular q-learning. The agent learns to forego a sequence of one or more measurements in predictable regions of the state space by repeatedly applying the same action. Their proposed method is found to requires fewer measurement step to reach the goal than the MDP baseline. However, it is only suitable for discrete state and actions spaces, and was only evaluated on grid-world problems. In this work, we show how action repetition and measurement skipping can be implemented in deep RL for continuous and image-based observation spaces.

DMSOA is a method that aims to improve the efficiency of RL. To this end, it is weakly related to other techniques to improve the sampling efficiency (19; 10; 18). The classic sample efficiency work, however, aims to reduce the overall number of training steps needed to learn a suitable policy, rather than reducing the measurement or decision steps made by the agent.

DMSOA utilizes concepts from the RL literature on dynamic frame skipping to repeatedly apply the selected control action (12). In DMSOA, however, the agent's measurement skipping policy is shaped by the intrinsic reward. Moreover, unlike frame skipping applications, which are concerned with processing speed not measurement costs, DMSOA does not have access to privileged extrinsic control rewards from intermediate steps. This making the problem more challenging.

In addition, DMSOA has a connection to the options framework (24), and particularly dynamic options (1). Similar to the options framework, at each decision point the DMSOA agent chooses to apply a sequence of actions that will transition the agent through multiple states. In DMSOA, the agent's policy selects a single control action and the number of times to apply the action in order to reduce its measurement costs while still achieving the control objective. Through the incorporation of measurement costs, the agent is able to learn how many times the action should be applied in order to arrive at the next meaningful state. For DMSOA, a meaningful state is one for which the information provided by it is greater than the cost to measure it.

# 3 Problem Setup

An AC-NOMDP is defined by $\langle \mathcal{S}, \mathcal{A}, \mathcal{O}, \mathcal{P}, \mathcal{R}_{ext}, \mathcal{R}_{int}, \mathcal{P}_{s_0}, \gamma \rangle$ where $S$ is the state space, $\mathcal{A} = \langle A_c \times A_m \rangle$ is the set of action tuples composed of control actions $a_c$ and binary measurement actions $a_m \in \{0, 1\}$ that specify if an observation of the next state is requested. The observation space $\mathcal{O}$ is related to $\mathcal{S}$ by the observation emission function $p(o|s', a)$ (more on this below). $\mathcal{P} : S \times A \times S \to \mathbb{R}$ denotes the transition probabilities, $\mathcal{R}_{ext} : S \times A_c \to \mathbb{R}$ denotes the extrinsic reward function, $\mathcal{R}_{int} : S \times A_m \to \mathbb{R}$ denotes the intrinsic reward function that encourages the agent to reduce the number of measurements it makes. The $r_{int}$ value is typically set to slightly outweigh the $r_{ext}$ value to achieve the balance between the need for information to solve to control problem and the cost of information. If $r_{int}$ is very large or very small relative to $r_{ext}$, the agent may never measure at the cost of solving the control objective or always measure and fail to reduce the observation costs.

The function $\mathcal{P}_{s_0} : \mathcal{S} \to \mathbb{R}$ denotes the probability distribution over the initial state and $\gamma \in (0, 1]$ is the discount factor. The observation emission function $p(o|s', a)$ specifies the probability of observing $o \in \mathcal{O}$ given the action $a$ in state $s$. Unlike the more general POMDP, the observation space in a AC-NOMDP is limited to $\mathcal{O} = \mathcal{S} \cup \{empty\}$, where $empty$ is the missing measurement of the environment. In this setup, the potential probabilities of $p(o|s', a_m = 1) \in \{0, 1\}$, with $p(o|s', a_m = 1) = 1$ if and only if $o = s'$ and $p(o|s', a_m = 1) = 0$ for all $o \neq s'$. In contrast, $p(o|s', a_m = 0) = 1$ if and only if $o = stale$.

The agent learns a policy $\pi(o) : O \to A$ that maps observations to action tuples. The initial observation $o_0 = s_0$ contains a fresh measurement of the environment. The control action selected by the agent is applied in the environment and the underlying state transitions according to $\mathcal{P}$. At each time step, the reward, $r_t$ is the intrinsic reward, $r_t = r_{(int,t)}$, if $a_m = 0$, otherwise the extrinsic reward, $r_t = r_{(ext,t)}$, is given in response to the state $s_t$ and control action $a_c$ selected by the agent. When the measurement action, $a_m = 1$, is selected, the agent receives a fresh measurement of the underlying state $o_{t+1} = s_{t+1}$ of the environment. Alternatively, when $a_m = 0$, the agent does not obtain a fresh measurement and the next action must be selected based the agent's internal mechanism, such as an internal memory or model.

174 The OSMBOA agent selects one control and one measurement action, $\langle a_{c,t}, a_{m,t} \rangle$ per time step,
175 whereas the DMSOA agent moves from decision point to decision point with a frequency less than
176 or equal to the environment's clock. At each decision point, the DMSOA agent selects a control
177 action and the number of times to apply it, $k$. The state is only measured on the $k^{th}$ application (e.g.
178 $\langle a_c, a_m = 0 \rangle_t, ..., \langle a_c, a_m = 0 \rangle_{t+(k-1)}, \langle a_c, a_m = 1 \rangle_{t+k}$).

179 The agent's objective is to learn a policy $\pi$ that maximizes the discounted expected costed return
180 which incorporates both the intrinsic and extrinsic rewards:

$$J(\pi) = \mathop{\mathbb{E}}_{a_t \sim \pi, s_t \sim P} \left[ \sum_t \gamma^t r(s_t, a_t) \right], \tag{1}$$

181 where $\gamma < 1$ is the discount factor. In this work, we focus on deep Q-learning based solutions (14)
182 combined with standard improvements for better convergence and stability (23; 25; 19). Although
183 this work examines problems with discrete action spaces, the proposed algorithms can be modified
184 for continuous action spaces.

## 4 Deep Dynamic Multi-Step Observationless Agent

186 The Deep Dynamic Multi-Step Observationless Q-learning Agent (DMSOA) for noiselessly observ-
187 able RL environments with explicit observation costs is presented in Figure 1. The framework has
188 three key components: the control policy $\pi_c : o \to a_a$ that maps the observation to a control action,
189 the measurement skipping policy $\pi_m : o, a_c \to k$ that maps the observation and selected control ac-
190 tion to $k \in \{1, ..., K\}$ the number of steps to apply $a_c$ to the environment, and the action-observation
191 scheduler. The action-observation scheduler applies the action pair $(a_c, 0)$ $k - 1$ times and collects
192 the intrinsic rewards $r_{int}$ from the environment. On the $k^{th}$ iteration, it applies the action pair $(a_c, 1)$,
193 records the extrinsic reward $r_{ext}$, and passes the new observation to $\pi_c$ and $\pi_m$. The extrinsic reward
194 is equal to the control policy reward for applying $a_c$ and arriving in the measured state after step $i = k$.
195 The intrinsic reward is $r_{int} \in \{0, c\}$, where $c$ is a bonus (ie. "cost saving") given to the agent when it
196 chooses not to measure. To ensure the agent is motivated to omit measurements whenever possible,
197 we set $c \geq r_{ext}^{max}$. The optimal setting of $c$ will depend on the application and the requirements of the
198 domain.

199 In this work, the policies are implemented as deep Q networks (DQN), however, other forms of policy
200 learning could be utilized. The agent's objective is to maximize the costed rewards $\sum_{t=0}^{\infty} \gamma^t r_t$. To
201 achieve this we learn parameterized value functions $Q_c(o; \theta)$ and $Q_m(o, a; \zeta)$ as feed-forward deep
202 neural networks. As described above, for an $m$-dimensional observation space and an $n$-dimensional
203 action space, $Q_c$ is a mapping from an $m$-dimensional observation to an $n$-dimensional vector of
204 action values. The function $Q_m$ is a mapping from an $m + 1$-dimensional observation-action to a
205 $K$-dimensional vector of measurement values. In the case of image data, each channel is augmented
206 with the action details. The argmax of each output indicates the action to apply and the number of
207 times to apply it.

208 During training, the experience tuples $(o_t, a_{(c,t)}, a_{(m,t)}, r_t, o_{t+1})$ are stored in a prioritized experience
209 replay buffer. To improve stability, target networks $\theta^-$ and $\zeta^-$ for $Q_c$ and $Q_m$ are copied from $\theta$ and
210 $\zeta$ every $\tau$ steps. The target for the control network is:

$$Y_i^{Q_c} \equiv r_t + \gamma \, Q_c \big( o_{t+1}, \text{argmax}_a Q_c(o_{t+1}, a; \theta_t); \theta_t^- \big). \tag{2}$$

211 For the same update step, the target for the measurement network is:

$$Y_i^{Q_m} \equiv r_t + \gamma \, Q_m \big( (o_{t+1}, a_{(c,t+1)}), \text{argmax}_a Q_m((o_{t+1}, a_{(c,t+1)}), a; \zeta_t); \zeta_t^- \big). \tag{3}$$

212 The corresponding losses are:

$$\mathcal{L}_i^{Q_c}(\theta_i) = \mathbb{E}_{(o_t, a_{(c,t)}) \sim \mathcal{D}} \big[ (Y_i^{Q_c} - Q_c(o_t, a_{(c,t)}; \theta_i)^2 \big], \tag{4}$$

213 and

$$\mathcal{L}_i^{Q_m}(\zeta_i) = \mathbb{E}_{(o_t, a_{(c,t)}, a_{(m,t)}) \sim \mathcal{D}} \big[ (Y_i^{Q_m} - Q_m((o_t, a_{(c,t)}), a_{(m,t)}; \zeta_i)^2 \big] \tag{5}$$

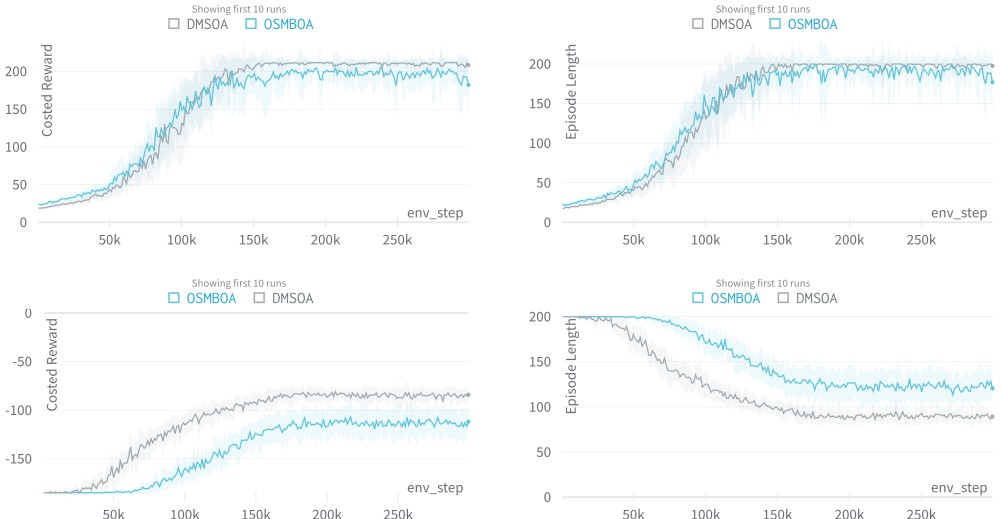

Figure 2: Mean and standard deviation of performance on Cartpole (upper) and Acrobot (lower). Left: costed reward and right: episode length. In both cases, DMSOA has a higher mean costed reward, and is superior in terms of the control objective (longer episodes on Cartpole and shorter episodes on Acrobot.)

## 5 Experimental Setup

In this section, we compare the performance of DMSOA to OSMBOA. In order to highlight the differences in the measurement behaviour of each method, we implement both with double DQN and a prioritized replay buffer. The hyper-parameters were selected via grid search with 3 random trials. For the evaluation, we report the mean and standard deviation of the reward during training and the observation behaviour of the best policy. Each agent is reinitialized with 20 difference seeds and trained on the OpenAI gym environments Cartpole, Acrobot, Lunar Lander and Atari Pong. The experiments were run on CentOS with Intel Xeon Gold 6130 CPU and 192 GB memory. In addition, a NVIDIA V100 GPU was used in the training of the Atari agent.

## 6 Results

Figure 2 shows the mean and standard deviation for each agent on the Cartpole and Acrobot environments. The aim in the Cartpole environment is for the agent to operate a cart such that a vertical pole remains balanced for as long as possible. The extrinsic reward is set to 1 and the intrinsic reward is set to 1.1. We truncate each episode at a maximum of 200 time steps. In the Acrobot environment, the objective is to apply torque to flip an arm consisting of two actuated links connected linearly above a target height in as few steps as possible. The agent receives an extrinsic reward of -1 or an intrinsic reward of -0.85 at each time step. Each episode ends after 200 steps or when the arm is successfully flipped over the line.

DMSOA learns a policy for both environments that produces a higher costed reward than OSMBOA. This indicates that DMSOA requires fewer measurements whilst carrying out the control policy. In addition, the standard deviation is lower indicating more stability across independent training runs. The episode length plots on the right show that DMSOA learned policies to keep the Cartpole upright longer and flip the Acrobot over the goal faster. Thus, DMSOA outperforms OSMBOA on all accounts.

Table 1: Ratio of steps with measurements to steps without measurements of the converged policy during training.

| Env. | DMSOA | OSMBOA |
|---|---|---|
| Cartpole | 1:1.27 | 1:0.37 |
| Acrobot | 1:0.45 | 1:1.03 |
| Lunar Lander | 1:1.56 | 1:0.33 |

The objective of the Lunar Lander environment is to fire the lander's rockets such that it lands squarely in the target area. In general, the results show shows

that DMSOA has significantly more successful landings than OSMBOA. Similarly to Cartpole and Acrobot, our analysis finds that DMSOA is more efficient in the number of decision steps and observations. More details on the environment along with performance plots are available in Appendix A Figure 5

Table 1 shows the ratio of the number of steps made without measuring for each measurement made. On Cartpole and Lunar Lander, DMSOA makes more than one step without measuring for each measurement step, whereas on Acrobot it makes an average of 0.5 non-measuring steps for each measuring step. This suggests that the dynamics of Acrobot are less predictable, causing DMSOA to measure more frequently on Acrobot. Interestingly, Acrobot is the only environment where OSMBOA does better than a 1:1 ratio.

## 6.1 Examination of Measurement Policies

Figure 3 shows the measurement behaviour of the best OSMBOA (left) and DMSOA (right) policies for Cartpole (top), Acrobot (middle) and Lunar Lander (lower) environments. Each row specifies a 1-episode roll-out of the best policy. Each column in the OSMBOA plots is the environment time step during the episode. For OSMBOA, the number of decision steps is equivalent to the number of steps in the environment. In contrast, each column in the DMSOA plots corresponds to a decision by the agent, with one or more environment time steps associated with it. In addition to highlighting the measurement efficiency, this also shows the decision efficiency. On Cartpole, DMSOA makes approximately 70 action selections (decisions) per episode of 200 environment steps (the mean steps per episode are shown in Figure 2.

For OSMBOA, an orange cell indicates that a fresh measurement of the environment and blue specifies that no measurement was requested at corresponding time step. In the case of DMSOA, the colour indicates the number of consecutive steps that were taken without a fresh measurement. Blue indicates that a measurement is made after the control action is applied once, yellow indicates that a measurement is made after the control action is applied twice and red indicates that a measurement is made after the control action is applied three times.

The distinct pattern in each plot suggests the different capabilities of each class of AC-NOMDP agent, along with the fact that each environment is unique in terms of its dynamics and complexity. The consistent measurement patterns for OSMBOA and DMSOA on Cartpole suggest that the environment has very regular dynamics. OSMBOA switches between selecting the next action from a freshly measured observation and selecting it from a stale observation. Alternatively, DMSOA learns to apply an action 3 times before measuring. This clearly demonstrates the potential of DMSOA to take more environment steps without measuring than OSMBOA.

Acrobot and Lunar Lander show much more complex measurement behaviour. For both OSMBOA and DMSOA on Acrobot during approximately the first 3/4s of the each episode they display a pattern of frequently measuring followed by briefly not measuring. Our analysis finds that both OSMBOA and DMSOA skip measurements while the arcobot is in the lower left region of the observation space. This is roughly where the momentum of the Acrobot shifts from heading way from the goal, back towards to the goal. In this area, it is deemed safe to apply torque back towards the goal without observing. In the last quarter of each episode, both agents take more steps without measuring. It is noteworthy that in most episodes OSMBOA takes significantly more steps without observing than DMSOA. However, this has a negative impact on the total number of action decisions made by OSMBOA on route to achieving to goal. This particularly visible in episodes 1 and 4. DMSOA does not to suffer from similar behaviour.

On the Lunar Lander environment both methods take few or no measurements near the end of the episode when the agent is close to landing. In addition, DMSOA repeatedly takes 2-3 steps before measuring at the beginning of each episode, whereas OSMBOA repeatedly measures early in each episode. Each method measures frequently during the middle of the episode as agent attempts to direct the lander safely towards the landing area. OSMBOA generally alternates between measuring and not measuring at each time step, whereas DMSOA typically takes many measurement steps followed by 1 to 2 steps without measuring before returning to measuring again. Similar to Acrobot, once OSMBOA estimates that it is on target to reach the goal it commits to never measuring again. When this estimate is erroneous, the leads to much longer episodes than necessary and the risk of crashing the ship.

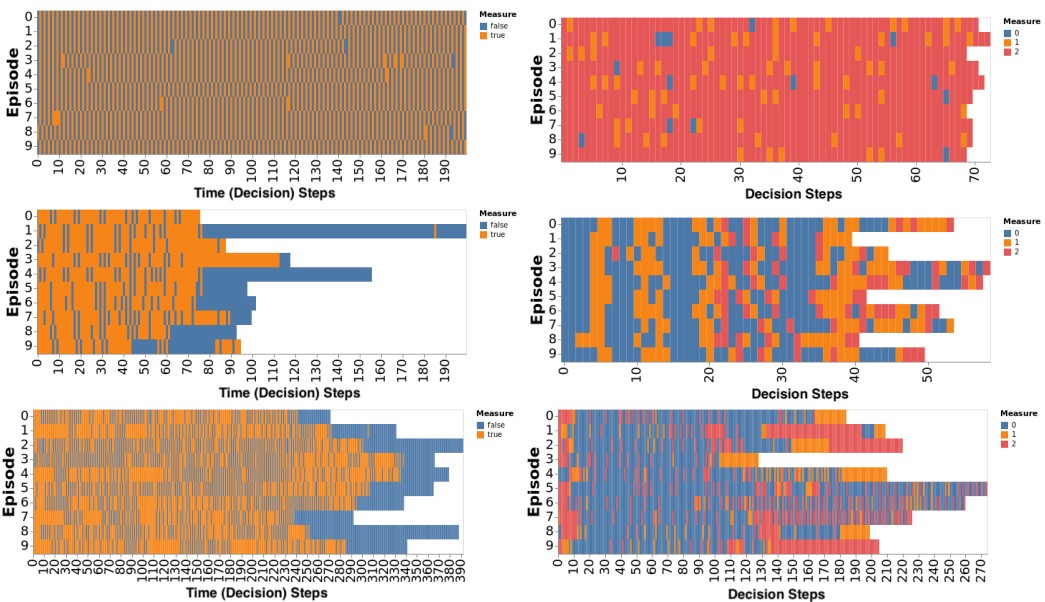

Figure 3: Measurement behaviour of the best OSMBOA (left) and DMSOA (right) policies. Top: Cartpole, centre: Acrobot, lower: Lunar Lander. For OSMBOA, blue indicates that no measurement was made and orange indicates that a measurement was made. In the case of DMSOA, blue indicates that a measurement is made after one step, orange indicates that a measurement is made after two steps, and red indicates that a measurement is made after three steps. This figure reveals the very distinct measurement behaviour between the two classes of AC-NOMDP agents.

## 6.2 Image-Based RL Results

The objective in the pong Atari game is to bounce the ball off of your paddle and past the opponents paddle into its goal (3). The action space is 6-dimensional including do nothing, fire, move right, move left, fire right and fire left. The observation space is a (210, 160, 3) image. In the case of OSMBOA, a 210 by 1 vector of ones or zeros is added to each channel to indicate if the observation is fresh or stale. The agent gets an extrinsic reward of 1 for winning a match and 0 for each intermediate step. Each episode is composed of 21 matches and the intrinsic reward is 0.001.

The results in Figure 4 show learns a policy to achieve a higher costed reward. In general, DMSOA learns to be a better Pong player and requires fewer measurements. Additional results showing DMSOA's advantage in terms of wins and intrinsic rewards can be found in Appendix A Figure 6.

From our analysis for the measurement policies of each agent, we found that both learn to measure less frequently when the ball is travelling away from their paddle. Alternatively, if the ball is near their paddle or the opponents paddle, each agent measures more frequently. Inline with the observations on Acrobot and Lunar Lander, when OSMBOA reaches a state from which it expects to win the match, it switch to not measuring for the remainder of the match. If the prediction is correct, it can achieve a greater reduction in measurements than DMSOA. If it is wrong, however, OSMBOA general loses the match. An erroneous prediction of this nature is particularly risky in a complex and dynamic environment.

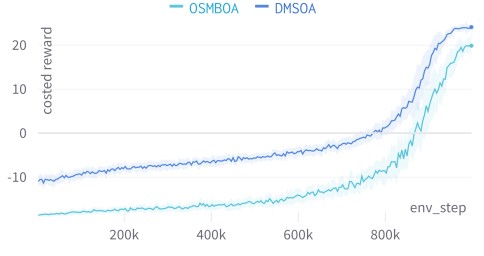

Figure 4: Mean and standard deviation of the costed reward on the Atari Pong environment. DMSOA learns a policy that for a better costed reward.

## 7  Discussion

The results indicate that DMSOA has a clear advantage over OSMBOA in terms of its convergence rate and the reduction in measurements and decision steps. We believe that the control action repetition capabilities of DMSOA improve its exploration of the environment and its understanding of the implications (positive and negative) of taking multiple steps without measuring. This helps it to quickly converge to good control and measurement policies. In addition, the fact that DMSOA's multi-step action sequences always ends with a measurement of the final state provides it with a good grounding from with to select the next control action. On the other hand, because the extrinsic reward for intermediate steps is not available, there is the potential for more noise in the reward signal for longer DMSOA action repetition trajectories. Due to the fact that OSMBOA is limited to one-step action, noise in the reward is less of a concern. Although, DMSOA appears to handle the noisy reward signal, future work should examine this in more detail.

For unshaped (or uniform) reward environments, such as Cartpole, Acrobot and Pong, setting the intrinsic reward is simple and the agent is insensitive to the value so long as it is slightly larger than the extrinsic reward. Alternative, the intrinsic reward requires fine tuning on environments with complex reward shaping such as Lunar Lander. As heuristic, we suggest starting the fine tuning from the mean of the extrinsic reward collected over multiple random walks in the environment.

In multiple environments, we found that OSMBOA commits to not measuring towards the end of each episode. This is surprising since if OSMBOA takes more than one step without measuring, it enters a partially observable state. This is akin to playing the game with its eyes closed. The agent is, thus, unaware if any unexpected event occurs. On Acrobot and pong, this resulted in it not achieving the goal, or it taking much longer than otherwise necessary. An example of this is seen in episodes 1 and 4 of the Acrobot plot in Figure 3.

## 8  Conclusion

In this work, we consider the problem of RL for environments where agent's decision making and measuring of the state of the environment have explicit costs, namely AC-NOMDPs. We provide the first survey of methods recently proposed for AC-NOMDPs. Building on the existing work, we propose DMSOA, an RL algorithm learns a control and a measurement policy to reduce measurement and decision steps. Our empirical results confirm the previously published results for OSMBOA on Cartpole, Acrobot and Lunar Lander, and show that OSMBOA is also capable on the more complex, image-based Atari Pong environment. However, we find that our proposed method DMSOA learns a *better control policy* than OSMBOA, and *requires fewer costly measurement and decision steps*.

This demonstrates the great potential to reduce measurement and decision costs associated with RL by allowing the agent to take control of its action and observation behaviour. We expect this to be a necessary capability of RL agents applied in many real-world applications. The next steps that we envision are developing more sophisticated loss functions for DMSOA, incorporating recurrency into the network to deal with time, expanding the analysis to additional methods and more realistic setting such as self-driving chemistry where observation can be costly and potentially destructive (2).

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

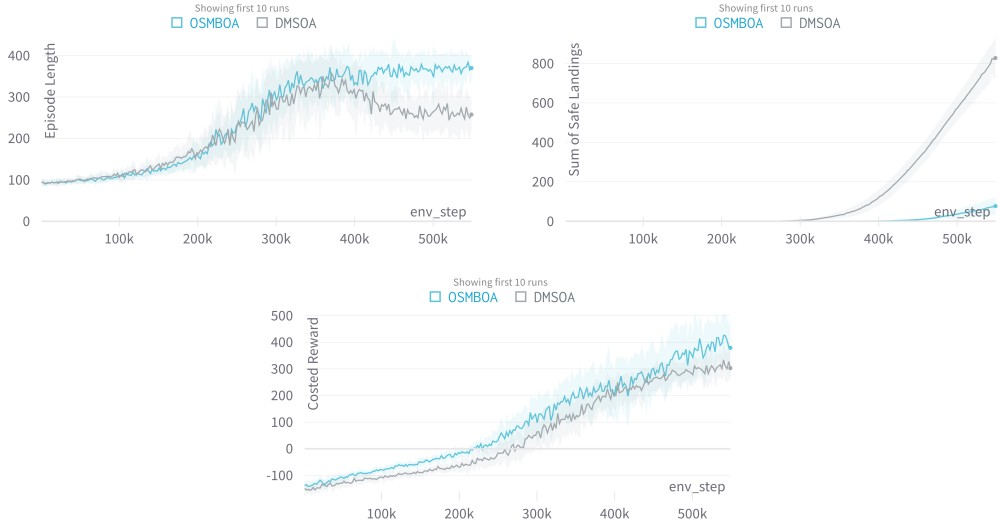

Figure 5: Mean and standard deviation of the performance on the Lunar Lander environment. Top left: episode length, top right: sum of the number of successful landings, lower: costed reward. DMSOA and OSMBOA achieve similar mean costed rewards. DMSOA, however, learns to successfully land the ship more frequently and in fewer steps.

# A    Appendix: Results

## A.1    Lunar Lander

The results for the Lunar Lander environment are presented in Figure 5. The Lunar Lander environment is a rocket trajectory optimization problem (7). The objective is to fire the lander's rockets such that it lands squarely in the target area. The fuel supply is infinite, but the best policy uses it sparingly. The environment has four discrete actions available: do nothing, fire left orientation engine, fire main engine, fire right orientation engine. The intrinsic reward is 0.1. The extrinsic reward is -0.3 for firing the main engine and -0.03 for side engines, the reward is also scaled by the lander's distance from the landing pad. Ten points are added to the extrinsic reward for each leg that is in contact with the ground, and an additional 100 points are added for landing, while 100 points are subtracted for crashing. The episode ends when the agent lands or crashes, or is truncated after a maximum of 400 time steps.

The plot on the top left in Figure 5 shows that mean episode length is longer for OSMBOA than DMSOA, and the top right plot shows that DMSOA has significantly more successful landings. This indicates that DMSOA learns a policy that quickly navigates the ship to a safe landing. The lower plot shows that OSMBOA has a slightly higher costed reward. As suggested by the first two plots, this is due to the fact that it takes longer to land and not because the policies is superior.

## A.2    Image-Based RL Results

The objective in the pong Atari game is to bounce the ball off of your paddle and past the opponents paddle into its goal (3). The action space is 6-dimensional including do nothing, fire, move right, move left, fire right and fire left. The observation space is a (210, 160, 3) image. In the case of OSMBOA, a 210 by 1 vector of ones or zeros is added to each channel to indicate if the observation is fresh or stale. The agent gets an extrinsic reward of 1 for winning a match and 0 for each intermediate step. Each episode is composed of 21 matches and the intrinsic reward is 0.001.

The results in Figure 6 show that DMSOA wins significantly more matches than OSMBOA (top left), achieves a higher costed reward (top right) and more intrinsic reward (lower). Thus, DMSOA learns to be a better Pong player and requires fewer measurements. Due the longer episodes and training times, a measurement behaviour plot similar to Figure 3 is not feasible within the confines of this paper.

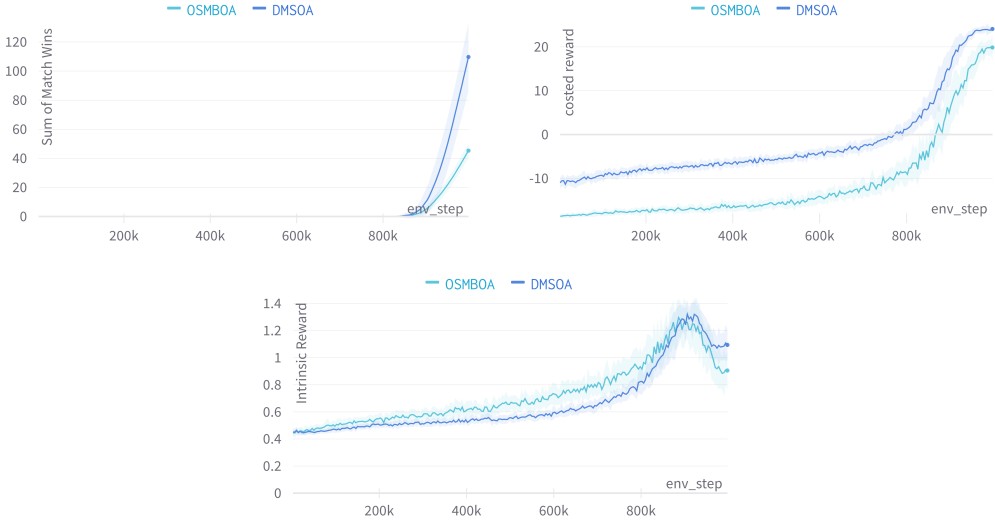

Figure 6: Mean and standard deviation of the performance on the Atari Pong environment. Top left: sum of the number of wins, top right: costed reward, lower: intrinsic reward. DMSOA learns a policy that wins more games with a better costed reward and fewer measurements.

From our analysis for the measurement policies of each agent, we found that both learn to measure less frequently when the ball is travelling away from their paddle. Alternatively, if the ball is near their paddle or the opponents paddle, each agent measures more frequently. Inline with the observations on Acrobot and Lunar Lander, when OSMBOA reaches a state from which it expects to win the match, it switch to not measuring for the remainder of the match. If the prediction is correct, it can achieve a greater reduction in measurements than DMSOA. If it is wrong, however, OSMBOA general loses the match. An erroneous prediction of this nature is particularly risky in a complex and dynamic environment.

