# OpenReview forum: "Dynamic Observation Policies in Observation Cost-Sensitive Reinforcement Learning"
_NeurIPS.cc/2023/Workshop/WANT — WANT@NeurIPS 2023 Poster_

### Official Review · Reviewer_ftGn · 2023-10-19
**Authors propose a deep reinforcement learning agent solving mdp with observations costs. Few baselines are compared to the proposed agent. Paper is clear. Not sure it fits the workshop.**

**Rating:** 5
**Confidence:** 4

**Review:**

The problem at hand in the proposed work is AC-NOMDP, a decision problem with actions that query environment measurements for a certain cost. The authors propose DMSOA, an agent that learns to repeat control actions for k steps until making a fresh measurement of the environment. They compare it to OSMBOA that instead learns to take a tuple (control action, do measure?) at every time step. Both agents maximize a cumulated reward that trades-off between control quality and cost of measurements.

In general the paper is well-written. I am convinced that DMSOA improves over OSMBOA even though DMSOA is a heavier model (with more parameters to fit) as it trains two Q functions: the control action one, and,the number of repetitions one.

It would be interesting to train a DQN agent on the non-costly MDP and then evaluating it against DMSOA by hiding the state, say every two steps.

One can see DMSOA as an RL agent training a policy that makes few measurements during evalutaion. In that sense, this work can be seen as saving resources. But the experimental section could be better (more baselines, more difficult tasks). Furthermore, DMSOA saves resources at evaluation time, but trains a heavier model. I am not sure I should recommend for acceptance. I am confident though in my understanding of the agent and of the problem.

typos: line 188 $a_a$ should be $a_c$, line 242 "show shows"

---

### Official Review · Reviewer_Bzuj · 2023-10-23
**Solving action-contingent, noiselessly observable Markov decision processes (AC-NOMDPs)**

**Confidence:** 3

**Review:**

This paper proposes a reinforcement learning algorithm to handle AC-NOMDPs. Unlike MDPs, the agent chooses whether to make a measurement at each step. Observations are said to be noisy, as the agent does not necessarily observe the environment. The proposed algorithm is  Deep Dynamic Multi-Step Observationless Agent (DMSOA). The agent receives an observation from the environment and outputs an action $a$ plus a number of step $k$ before the next measurement.

The problem is well stated and I really like that the algorithm outputs a number of step before the next decision is made. I think it's a desirable property to be able to choose an action for a given number of steps instead of choosing an action at each step for many decision-making problems. Furthermore, I like the results provided in Section 6.1 because they highlight this reduction in decision-making costs.

However, there are a few grey areas. The choice of parameter $K,$ which determines the output size of $Q_m(.,.;\zeta)$, is not detailed, either in the results or in the appendix. This may be a good point to add some details about it. Indeed, in Section 6.1 on figure 3, we can see that the agent can wait 3 steps before making a measurement. It is not clear whether this is due to the choice of K or whether it is the agent who has chosen never to go beyond three steps. I have also noticed that in several RL papers, some use cases are presented in the introduction, but are rarely present in the experimental section. I think it would be beneficial for the whole RL community to try testing agents on some of these use cases to really show that the RL algorithm can be used beyond toy problems.

---

### Official Review · Reviewer_jJjy · 2023-10-24
**This paper address problem of COST-SENSITIVE REINFORCEMENT LEARNING (RL). Its important scenario in specific but real-world application of RL.**

**Confidence:** 3

**Review:**

This paper address problem of COST-SENSITIVE REINFORCEMENT LEARNING (RL). Its important scenario in specific but real-world application of RL  such as  materials design, financial trading[1] and  complex robotics.

Authors use special case  action-contingent, noiselessly observable Markov decision processes (AC-NOMDP).
It is worth to mention that the authors did extensive survey of related works. They propose a novel method called "Deep Dynamic Multi-
Step Observations Agent" to improve convergence, and reduce measurement and decision steps in cost-sensitive RL environment. In contrast with the baseline methods, DMSOA selects a control action and how many apply it before measure the next state.

The manuscript is  clearly written and has and illustrations. However, the data on Figure 3 could also be presented differently, maybe as as table with final measurement statistics.

For a workshop level paper I don't see any critical limitations. Overall, this is promising work that address rare setting of RL. It would be interesting to see application of DMSOA  in real problems, such as material science and complex control systems.

[1] Y. Zhang, P. Zhao, Q. Wu, B. Li, J. Huang and M. Tan, "Cost-Sensitive Portfolio Selection via Deep Reinforcement Learning," in IEEE Transactions on Knowledge and Data Engineering, vol. 34, no. 1, pp. 236-248, 1 Jan. 2022, doi: 10.1109/TKDE.2020.2979700.

---

### Meta-Review · Area_Chair_wH32 · 2023-10-26

**Recommendation:** Accept (Poster)
**Confidence:** 4

**Metareview:**

The authors propose to learn neural policies for RL that learn to potentially repeat the same action for multiple time-steps to reduce the computational cost. Reviewers are generally positive about the submission. There is a concern from one reviewer that only test time is improved whereas train time is more computationally demanding, but I still think that the paper remains relevant for the workshop.

---

### Decision · Program_Chairs · 2023-10-28

**Decision:**

Accept (Poster)

**Comment:**

We thank the authors for their time and contribution to WANT and we are pleased to share that after the reviewing process the paper has been accepted. Congratulations! We encourage the authors to consider reviewers' feedback for the improvement of the camera-ready version. We hope to see you in person at the workshop and brainstorm on efficient training research together!